# Effect of the Composition of Hybrid Sands on the Change in Thermal Expansion

**DOI:** 10.3390/ma15176180

**Published:** 2022-09-05

**Authors:** Filip Radkovský, Martina Gawronová, Václav Merta, Petr Lichý, Ivana Kroupová, Isabel Nguyenová, Šimon Kielar, Martin Folta, Josef Bradáč, Radim Kocich

**Affiliations:** 1Department of Metallurgical Technologies, Faculty of Materials Science and Technology, VSB-Technical University of Ostrava, 17. Listopadu 2172/15, 708 00 Ostrava, Czech Republic; 2Brembo Czech, s.r.o. Na Rovince 875, 720 00 Ostrava, Czech Republic; 3Department of Production, Logistics and Quality, ŠKODA AUTO University, 293 01 Mladá Boleslav, Czech Republic; 4Department of Mechanical and Electrical Engineering, ŠKODA AUTO University, 293 01 Mladá Boleslav, Czech Republic

**Keywords:** discontinuous thermal dilatation, silica sand, CERABEADS, synthetic ceramic sands, phase transition, foundry moulding mixture

## Abstract

In the foundry industry, silica sands are the most commonly used type of sands for the production of sand foundry moulds using various types of binders. Their greatest disadvantage is their significant volume changes at elevated temperatures, which are associated with the formation of many foundry defects from stress, such as veining, and thus have a direct influence on the final quality of the casting. In the case of non-silica sands and synthetic sands, the volume stability is more pronounced, but this is accompanied by a higher purchase price. Therefore, a combination of silica sand and synthetic sand CERABEADS is considered in order to influence and reduce the thermal expansion. The hybrid mixtures of sands, and their most suitable ratios, were evaluated in detail using sieve analysis, log W and cumulative curve of granularity. It was found that the addition of 50% CERABEADS achieves a 32.2% reduction in dilatation but may increase the risk of higher stresses. The measurements showed a significant effect of the granulometric composition of the sand on the resulting thermal expansion, where the choice of grain size and sorting can achieve a significant reduction in dilatation with a small addition of CERABEADS.

## 1. Introduction

Foundry moulds and cores consist of three main components: a foundry sand, a binder (which binds the sand grains together) and additives to improve the properties of the moulds and cores and the quality of the final product, the casting. The main ingredient is the sand, which ranges from 84 to 98% by weight in the moulding compound. There are many requirements that these materials must meet to be suitable for use in foundry applications. They must be chemically inert to molten metals and compatible with binder systems, dimensionally and thermally stable at elevated temperatures, have as little gas evolution as possible on heating and have consistent composition, pH and suitable granulometric properties [1].

Foundry sands consist of 2–3 major and several minor grain size fractions in the range of 0.063 to 1 mm. The distribution of the individual fractions largely determines the average grain size of the sand. However, it is mainly the consistency of the particle size distribution that is critical for achieving the maximum result when using sand in a moulding mixture. Its role is well known; the grain size distribution of the sand determines the amount of binder required, the compactibility and permeability of the moulds and other mechanical and technological properties [2]. In addition to the physical and chemical properties of the sands, economic criteria are also very important for the use of a given type of sand. This mainly means purchase price and availability. From this point of view, silica sand is the most widely used in the foundry industry. This comes from the fact that silica is the most widely occurring material found in nature in an optimal shape, moreover, its resistance at elevated temperatures is suitable for standard use in the foundry industry [3]. Speaking of silica sand as a moulding material, it is defined as mineral quartz. It is required to contain a minimum of approximately 95% SiO_2_, the exact composition of which is determined for each deposit and batch [2,3].

However, silica sands have a number of technological and hygienic disadvantages. The hygiene disadvantages are mainly represented by the lung disease silicosis. The most significant technological disadvantage in the use of silica sands, apart from their acid nature, is their behaviour at elevated temperatures. In addition to the formation of fayalite on contact with the iron alloy melt, which reduces the heat resistance of the basic sand, discontinuous dilatation also occurs. This is due to the phase transition of SiO_2_, which takes place at about 573 °C when the α-quartz is converted to β-quartz. Changes in crystallographic structure lead to changes in specific density, so that grain expansion occurs [1]. As reported, for example, by [2], in the studied samples the dilatation varied with temperature change and the maximum dilatation measured was up to 2.25% [2]. As a result of this phase transition, the silica sands change their density, and this results in volumetric changes that cause stress defects in the moulds which negatively affect the surface quality of the casting. These defects can be prevented by certain technological steps or mechanically removed after the casting has solidified. However, these steps increase the cost and length of the manufacturing process [3,4,5].

An important factor is therefore generally the dimensional stability of the moulds and cores, which is highly dependent on the properties of the base sand and the binder used. Dimensional stability is also influenced by the impurity content of the sand. Silica sands tend to contain various mineral contaminants such as potash, alkali feldspars and soda, which may occur as separate particles or as a layer on the surface of the silica grains. These contaminants can significantly reduce the adhesion between the binder and the sand grain, as well as reduce the cohesion and strength of the binder bridges [6,7,8].

The relationship between grain size distribution and thermal properties such as thermal expansion is commonly studied in foundry research on unbonded and bonded sand samples [9,10,11,12,13]. From the mould/metal interface, the mould layers heat up to a certain temperature range due to the relatively low thermal conductivity of the porous mould. Therefore, the rate of expansion varies depending on the distance from the casting. This can lead to mould deformation and can cause a range of surface defects in the casting such as expansion scabs, veinings and others [1]. Previous authors have found a clear link between increasing grain size and increasing thermal expansion [11,12]. However, this correlation is generally explained purely on the basis of the difference in physical size fractions and neglects the state of chemical composition. The thermophysical properties of moulding materials play an important role in the solidification process of castings. Attributes such as specific heat determine the cooling capacity of a mould, e.g., the amount of heat it can absorb and dissipate from the molten metal to the surroundings [2,14,15].

Thiel et al. [11] investigated the effect of various factors such as grain size, grain shape, granulometric distribution (sieve analysis), chemical composition and density on the thermal expansion of silica sands. They studied four different size ranges of silica sand and the correlation found indicates that the finer the grain, the lower the maximum expansion. According to the thermal expansion behaviour of silica sands, it is also evaluated that with higher silica content, these sands have higher expansion.

Interpretation of measurement results related to granular materials is always challenging because the system consists of grains in contact. Several factors influence the results for the observed phenomena, which are difficult to distinguish. In addition, in moulds and cores, the sand grains are bonded together by a binder. The types of binders vary widely, where the choice ranges from organic binders to inorganic ones, and there is a difference in the strength and flexibility of the bond formed [16]. When the temperature increases, the binder medium counteracts the expansion. Then, it is difficult to separate the effect of the binder and the base sand. For this reason, it is crucial to study unbonded sand grains when the aim is to clarify the origin of certain phenomena [17].

Previous research has shown that the veinings are the result of tensile stresses acting at the mould/metal interface due to dilatation of the sand in the subsurface layers of the mould. This situation is due to the phase transition of the silica sand at a temperature of about 573 °C. There is a non-uniform temperature distribution in different areas of the sand at different distances from the heat source—the molten metal—which then creates irregular thermal expansion and associated stresses. When the forces on the mould or core surface are high, the liquid metal has the opportunity to penetrate these gaps in the cracking mould material [18]. The dilatation of each grain (microdilatation) manifests itself as macrodilatation of the mould or core. The value of macrodilation then depends on the degree of freedom of microdilation of each grain (additives, binder content, grain shape, compaction degree). The highest value of dilatation is found in monofraction sand with rounded grains. In [18], the most effective approach to prevent stress defects is to replace the sand with another sand with lower thermal expansion (deformation) values, without phase transformations [19,20,21].

Due to the above disadvantages, silica sands are not suitable for use in all foundry processes. Therefore, a variety of non-silica sands are used in foundries. One of the many types of non-silica sands is the synthetic aluminosilicate sand CERABEADS. This refractory sand has a number of advantageous properties, in particular high thermal absorption and very low thermal expansion, thus ensuring high surface quality castings. However, the purchase cost of these sands is considerably high compared to silica sands [4].

The aim of the experiment was to create a hybrid mixture of silica sand with aluminosilicates CERABEADS, assuming a reduction in thermal expansion of silica sand and hence a reduction in the risk of occurrence of casting defects, and therefore a reduction in the time and cost of casting production and so an increase in its final quality. As the main material investigated, BG 27 silica sand was chosen with a granulometric composition corresponding to the sands used for core production, whose high chemical purity (above 99.5%) increases the predisposition to the formation of stress defects, especially veining. CERABEADS (NCB) aluminosilicate sands, namely NCB 650, NCB 950 and NCB 1450, with different granulometric composition and mean grain size, were then added to the hybrid mixtures in a 0–50% ratio.

## 2. Materials and Methods

### 2.1. Sand Choice

As a representative of natural sand, Biala Góra foundry silica sand marked as BG 27 with a mean grain size of AFS: 60 was selected. The main specification was high chemical purity (<99.7% SiO_2_), angular grain shape and optimal granulometric composition for core mixtures. Silica sands are economically affordable and therefore very widespread. However, they are characterized by a very discontinuous dilatation curve and a high coefficient of thermal expansion, i.e., there is an increased risk of stress defects.

The synthetic sand used for the experiment was the aluminosilicate sand brand CERABEADS, marked as NCB, with three AFS mean grain sizes according to the supplier, namely NCB 650 (AFS: 65), NCB 950 (AFS: 95) and NCB 1450 (AFS: 145). The production of this type of sand is based on spraying ceramic slurry into the hot chamber of the kiln to form individual balls/grains of sand. It is significantly more expensive compared to silica sands. However, it has a very low coefficient of thermal expansion and its dilatation curve is smooth. This means it contributes significantly to reducing the occurrence of stress defects.

### 2.2. Sample Preparation

Silica sands are characterized by high thermal expansion, which is a common cause of stress casting defects. Based on the theoretical assumption of a possible reduction in the dilatation of the sand by mixing silica sand with a minimum thermal expansion sand, it was chosen to use CERABEADS (NCB) sand up to a content of 50%. All the investigated sands were used as 100% reference samples and also hybrid mixtures of BG 27 with the addition of NCB 650, NCB 950 and NCB 1450 were prepared at 10%, 20%, 30%, 40% and 50% NCB. The marking of the samples and the content of each component can be seen in Table 1. At least 3 samples of each initial sand and hybrid mixture were prepared and measured every time, and the resulting values were then averaged.

### 2.3. Sand Evaluation

#### 2.3.1. Powder Bulk Density and Sieve Analysis

The powder bulk density of both pure and hybrid sands was determined according to American Foundry Society standard procedure AFS 1131-00-S by freely pouring the dried sample into a container of well-defined volume (graduated cylinder), where the sand sample was shaken to a constant height by vibration. The volume of the compacted sample was read using a ruler and the sample was weighed. The bulk density was then determined by the ratio of the sample weight to the actual sample volume in the container. At least 3 measurements were performed for each hybrid mixture.

A detailed overview of the granulometric composition was carried out by sieve analysis. Firstly, the determination of clay wash analysis according to ASTM C117-13 was performed to remove dust and other small particles smaller than 0.02 mm from the sands. The principle of measurement was based on the different sedimentation rates of particles of different sizes. The weight of each sample for sieve analysis was 50 g. The measurements were performed on a laboratory sieving machine (LPzE-2e, Multiserw-Morek, Poland) equipped with a set of sieves with graduated mesh size (mesh in mm: 0.710, 0.500, 0.355, 0.250, 0.180, 0.125, 0.090, 0.063, 0.020 pan) according to ASTM E11. The sieving time was set to 10 min. The weight of the fractions captured on each sieve was then expressed as % relative to the weight of the original sample. The American Foundry Society (AFS) mean grain size was determined by calculating the percentage of sand fraction captured on each sieve using a multiplier.

From the results of the sieve analysis, the cumulative curves of granularity were determined and the homogeneity degree *S* according to Equation (1) and the criterion of the grain size distribution probability *log W* according to Equations (2) and (3) were calculated [22].
(1)$$S=\left(d_{75}/d_{25}\right)\cdot 100\left(\%\right),$$
where *d*_25_ and *d*_75_ are the mesh diameters corresponding to 25 and 75% of the total weight of the sand without particles smaller than 0.02 mm.
(2)$$N_{i}=\left(m_{i}/m_{p}\right)\cdot 100\left(\%\right),$$
(3)$$logW=100\cdot log\left(100\right)-\sum N_{i}log\left(N_{i}\right),\;$$
where *N_i_* is the fraction captured on the sieves (%), *m_i_* is the weight of the sample on the sieves (g) and *m_p_* is the weight of the sample after determination of the clay wash analysis (g).

#### 2.3.2. Thermal Dilatation of the Sands

The aim of the measurements was to observe the behaviour of the material at certain temperatures from the shape of the dilation curve and to monitor the phase transitions. The measurements of the thermal expansion of the sands were performed using a dilatometer (DIL 402/C, Netzch, Selb, Germany) equipped with corundum components (rod, pads, container). Before the measurement, a correction was made using a corundum correction sample of calibrated size 10 mm inserted into a cylindrical corundum container with plugs. Samples of the sand for measurement were prepared by pouring into the container cavity and then compacting with 3 light strokes of a laboratory spoon. The height of the measured sand sample in the plug container was within 10 mm ± 0.05 mm. The measurements were carried out in a temperature range of 25 °C to 1130 °C under an inert atmosphere (argon 6.0, flow rate 100 mL/min). The temperature rise was set to 15 °C/min. At least 3 measurements were performed for each hybrid mixture.

## 3. Results and Discussion

### 3.1. Sand Characterisation

Two types of basic sand with different grain size, shape and granulometric composition were chosen to verify the effect of grain size on the dilatation curve. Silica sand BG 27 was characterised by high chemical purity and angular grain shape (Figure 1a). NCB 650 was declared to have a similar mean grain size to the BG 27 silica sand, so the resulting granulometric composition of the hybrid mixture should not be significantly affected in this case. The remaining NCB 950 and NCB 1450 sands were characterised by finer granulometry. This synthetic sand was also characterized by a rounded grain shape compared to the silica sand used, as can be seen in Figure 1b–d.

### 3.2. Powder Bulk Density and Sieve Analysis

Silica sands of high chemical purity are generally around 1.5–1.7 g/cm^3^ [23,24]. The bulk density of CERABEADS is reported to be around 1.7 g/cm^3^ [23]. The bulk density of the sand depends significantly on its chemical composition as well as on the shape and size of the individual grains. In general, the smaller and rounder the individual grains, the more the bulk density increases. Figure 2 compares the average bulk density values of NCB, BG 27 and their mixtures. It can be noticed that the difference in bulk density values between BG 27 100% and NCB 650, 950 and 1450, is minimal. This is a positive result which shows that the sands can be easily mixed, especially without sedimentation of the different types of sands. Generally, during moulding, sands with higher density (higher mass) can segregate and negatively affect the properties of the mould or core in the cross-section. Materials with similar bulk density do not normally show this segregation during moulding and can thus be considered as well blendable. The bulk density of silica sand BG 27 was 1.66 g/cm^3^, while for CERABEADS it ranged from 1.75 to 1.76 g/cm^3^ depending on the granulometry. It has been confirmed that a slight increase in bulk density occurs by successive mixing with sands with smaller grain sizes. Specifically, the addition of 50% CERABEADS resulted in a 5.4% increase over BG 27 for NCB 650 and NCB 950, and a 6.0% increase for NCB 1450.

The sieve analysis provided a comprehensive overview of the granulometric composition of both pure sands and the development of the granulometry of hybrid sand mixtures. The resulting weights of the fractions captured on the sieves, the value of the mean grain d_50_, then d_25_ and d_75_, the homogeneity degree S and the grain size distribution probability criterion log W are presented in Table 2, Table 3 and Table 4. Table 2 compares the results of BG 27 100% with the addition of NCB 650, Table 3 with the addition of NCB 950 and Table 4 with NCB 1450.

In all three cases of NCB sand addition, regardless of type, there is a significant decrease in the mean grain size d_50_ depending on the amount of NCB added, i.e., a refinement of the granulometric composition. The most significant difference in granulometric composition is observed in the results (Table 2, Table 3 and Table 4) for the addition of 50% NCB. Especially for the NCB 1450 type, where there is a 30% refinement of the mean grain size between the initial BG 27 100% and the NCB 1450 50%. In the case of NCB 650 50%, there is only a 9% refinement of the mean grain size.

Since the natural sand used as a base was silica sand BG 27, it is also possible to evaluate the homogeneity degree S, for which the more homogeneous the sand is, the more the S value approaches 100%. NCB synthetic sand with a smaller mean grain size d_50_ was added to the hybrid mixtures, which is characterized by high homogeneity compared to natural sands (almost 50% of the sand is concentrated on one fraction, usually the one closest to the total d_50_), as can be seen from the sieve analysis for NCB 100%. This has resulted in an increase in the proportion of sand trapped on several different sieves of different sizes, i.e., a so-called further ‘dilution’ of the already heterogeneous natural sand, and in particular an increase in the volume of finer grains on the sieves. Thus, the homogeneity of the hybrid mixtures (S value closer to 0) was significantly reduced compared to BG 27 100% by 5.7% in the case of NCB 950 50% and by 10.2% in the case of NCB 1450 50%. The mixtures thus became more heterogeneous due to the blending. The exception here is the NCB 650 50% mix, where on the contrary, there was a slight increase in the homogeneity of the mix by 5.6%, i.e., a decrease in heterogeneity, due to the use of NCB 650 sand with a very similar granulometric composition to BG 27 100%, where most of the sand grains were trapped on 0.250–0.125 mm sieves.

The values of *d*_50_, *d*_25_ and *d*_75_ were obtained from the cumulative curves of granularity shown in Figure 3 and Figure 4. The cumulative curves of all the 100% sands and hybrid mixtures used can be seen in Figure 3, where a gradual transition from polyfractional mixtures towards more monofractional ones can be observed due to the different percentage of the addition of synthetic NCB sands. Figure 4a shows the cumulative curves of only pure 100% sands without blending. The most striking difference here is between the linear curve for BG 27 100%, which is more polyfractional in nature, and the curve for NCB 1450 100%, which is the steepest, and thus shows the more monofractional nature of the sharply sorted sands. Figure 4b shows the cumulative curves for mixtures with NCB 650, Figure 4c shows the curves for mixtures with NCB 950 and Figure 4d shows the curves for mixtures with NCB 1450. In all three cases, both a change in the mean grain size with increasing additions of NCB sands can be observed, as well as a change in the curve to a more linear (polyfractional) curve in the case of NCB 950 50%, which corresponds to the calculated *log W* value.

The more the criterion of the grain size distribution probability log W approaches 0, the more sharply the sand is sorted, i.e., it is a monofraction [12,22]. This means that the largest part of sand grains is concentrated on one or maximum two sieves. Conversely, the more the log W value approaches 100, the more regularly the same amount of sand is distributed on all sieves, i.e., it is a polyfraction. The measured results show that the log W value of 64.0 for the silica sand BG 27 100% is closer in nature to polyfraction. This means that it has far more sand grains more evenly distributed in several different fractions of different sizes. In contrast, the NCB 650 100% with a log W value of 56.3 and NCB 950 100% with a log W value of 60.5 have the highest concentration in fractions around d_50_, and the NCB 1450 100% with a log W value of 37.0 contains almost no, or only minimally, fractions other than those belonging to d_50_ (Figure 4a). Thus, NCB 1450 100% can be characterized as a sharply sorted sand with a character close to monofraction. However, the gradual addition of these sands to BG 27 up to 50% NCB has the opposite effect to that of approaching monofraction. While only a slight decrease in the log W value of the hybrid blend NCB 650 50% was observed, namely by 1.7%, it can be concluded that the addition of synthetic sands of very similar granulometric composition did not significantly affect the resulting granulometric composition and did not change the character of the sands (Figure 4b). In contrast, the hybrid blends NCB 950 and NCB 1450 increased the log W value, i.e., the polyfractional character of the sand, by 9.5% and 8.3% in the case of NCB 950 50% and NCB 1450 50%, respectively. This is due to the addition of a very fine monofractional sand, where the largest proportion of grains is in completely different fractions than in the case of BG 27 100%. This increases the amount of sand grains trapped on multiple meshes of different sizes, and changes the character of the basic sand to a more polyfractional one with a more even representation of the different fractions (Figure 4c,d).

Mixtures with polyfractional distribution of sand grains usually achieve higher mechanical strengths, but are characterised by higher specific surface area and therefore higher binder consumption and lower permeability of the mixture [22]. This may be the source of increased raw material costs or bubble-type foundry defects. Monofraction mixes are less demanding on excess binder while maintaining sufficient strengths, thus saving binder. In contrast, they tend to allow higher penetration of metal into the intergranular spaces due to their higher porosity and are thus more vulnerable to foundry defects such as burning, flash and veining or increased surface roughness. This is due to the absence of fine sand grains filling the intergranular spaces.

### 3.3. Thermal Dilatation of the Sands

The performed experiment provided a comprehensive overview of the one-directional thermal expansion for the individual sands in the pure state, as well as the thermal expansion of hybrid mixtures with different CERABEADS content in silica sand. The evaluated mixtures showed different granulometric compositions compared to the sands in their native state. The graphs below show the dilatation curves of all mixtures compared in the experiment. Figure 5a shows the dilation curves for the sands used in their original state, i.e., 100% content. The first fact that cannot be overlooked on the graph is the significant difference in the shape of the curves. The NCB sands are characterized by a smooth, linear dilation curve with a slight increase. The measured thermal expansion values were 0.56%, 0.36% and 0.46% for NCB 650 100%, NCB 950 100% and NCB 1450 100%, respectively. In contrast, the silica sand BG 27 100% exhibited a significantly discontinuous dilatation curve. This is caused by changes in crystal lattice during β-SiO_2_ to α-SiO_2_ phase transition, which took place at 556.4 °C for the sample. As a result of phase transition, silica sand changes its mass density that leads to volume changes. Compared to the dilatation of NCB sands, the resulting dilatation of 1.49% for BG 27 100% is 2.6× to 4.5× greater. From Figure 5b, it can be seen that as the percentage of NCB increases, the percentage of expansion decreases for all blends below the dilation value of BG 27 100%, i.e., below 1.49%. The lowest dilatation for the hybrid mixture was achieved in the case of NCB 1450 50% mixture, namely a dilatation of 1.01%, which is 32.2% lower dilatation than that of pure silica sand. In the case of silica sand, high temperatures during casting lead to polymorphic transformations, which has the significant disadvantage of discontinuous thermal dilatation. This is due to the reversible phase change β ↔ α of SiO_2_ around 573 °C. This transformation takes place within seconds and is accompanied by a slight change in density and linear dilatation of the mixture. The inhibition of free dilatation leads to a sharp increase in stresses in the mould and its expansion, which leads to a decrease in the accuracy of the casting and to the formation of a number of foundry defects from thermal stresses (scabs, veinings). The value of the expansion varies depending on the type of basic sand used, the binder, the compacting of the mould or additives allowing stress relaxation. Thermal expansion of silica is worse in iron alloy castings, especially when using high-purity silica sands. The above characteristics, in particular the discontinuity of the thermal expansion curve, make silica sands significantly different from other sands used in the foundry industry.

One of the main reasons for using non-silica sands, apart from suppressing the acid reaction of silica sands at high temperatures and increasing the heat resistance of moulding and core mixtures, is to suppress a number of foundry defects caused by the significantly higher thermal expansion of silica sands. In general, non-silica sands have significantly lower thermal expansion values, with overall rates typically less than 1%.

Figure 6a–c show the detailed dilatation curves for different types of NCB at different quantities compared to the initial BG 27 100%. When 50% NCB was added to the silica sand, the resulting dilation was 1.11%, 1.19% and 1.01% for NCB 650, NCB 950 and NCB 1450, respectively. Thus, a reduction of up to 32.2% in dilatation was achieved by the addition of 0–50% NCB compared to the initial state (for the addition of NCB 1450, and 25.5% and 20.1% for the addition of NCB 650 and NCB 950, respectively). There is still a noticeable discontinuity in the dilatation curves of the NCB mixtures, where the effect of the change in the β ↔ α SiO_2_ phase transition is significantly noticeable. The phase transition for all samples took place at 555.38 °C ± 1.5 °C. Thus, there is a very noticeable effect of silica sand as there was no significant shift in this temperature. The above results are also reflected in the decreasing coefficient of thermal expansion (Figure 7), with BG 27 again reaching the highest values and NCB 950 reaching the lowest values (by 70.14%) in its original state with the lowest error range between samples. In the case of hybrid mixtures with 0–50% NCB, the highest reduction in the coefficient of thermal expansion compared to BG 27 100% was achieved for the mixture NCB 1450 50%, namely by 35.26% and 26.54% for the mixture NCB 650 50% and 22.08% for the mixture NCB 950 50%, respectively. In contrast, samples of the NCB 1450 50% mixture showed the highest error range value of all measured mixtures, with some measurements showing thermal expansion coefficient values almost identical to the NCB 950 50% mixture. Exact results can be seen in Table 5. The results show that neither group of hybrid mixtures showed a linear trend with increasing percentage of NCB sand. This effect was caused by the gradually changing granulometric composition and in particular by a combination of changes in the homogeneity of the mixtures, as expressed by the sorting coefficient S, and changes in the polyfractality of the sands. The changes in granulometric composition are explained in more detail in Section 3.2 and their relation to thermal expansion below.

Although the hybrid mixture NCB 1450 50% showed the lowest final dilatation of the studied sand mixtures, on the other hand, it showed a much steeper increase in dilatation during the phase transition to α-SiO_2_. Thus, a higher stress from inhibited dilatation in the mould or core and a higher risk of stress defects can be expected if moulds or cores without protective coating are considered. From this point of view, the hybrid sand NCB 650 50% appears to be preferable, which, although it achieved a dilatation of 9.9% higher compared to NCB 1450 50%, showed a lower increase at 556.4 °C. The mixture NCB 950 50% did not show significantly improved properties compared to mixtures with lower NCB content, and this option would be less advantageous.

The resulting dilatometric curves (Figure 5b) further show that, overall, the most significant reduction in dilatation, especially in the area of phase transition to α-SiO_2_, was achieved in the case of the addition of NCB 650. In the case of 10% NCB 650 addition, there was no noticeable difference, but already from 20% NCB 650 addition there was a significant decrease in dilation, compared to fine-grained NCB 950 and NCB 1450. The decrease in dilatation for the 20% NCB addition was 14.1% for NCB 650, compared to 8.1% for NCB 950 and 2.7% for NCB 1450. Changes in the coefficient of thermal expansion showed the same trend (Figure 7). This result is consistent with the result of sieve analysis, whereby the NCB 650 was closest to the silica sand BG 27 in terms of mean grain size d_50_, while its granulometric composition increased the homogeneity of the hybrid mixture, i.e., increased the percentage of sand grains trapped on fractions around d_50_ (Table 2). Although the addition of NCB 950 from the 30% level onwards led to a gradual reduction in thermal expansion in the phase transition area, the resulting dilatation values differed only minimally. Thus, a higher addition of this sand did not show a significant effect on reducing the overall dilatation of the hybrid mixture. Gradual addition of NCB 1450 always resulted in a consistent reduction in total dilatation, and the hybrid mixture with NCB 1450 50% also achieved the lowest total dilatation, despite a steeper increase in the phase transition area. The reason for this can again be deduced from the results of the sieve analysis (Table 4), whereby the gradual mixing of NCB 1450 into the silica sand resulted in a more uniform distribution of the sand grains into more fractions (increasing the heterogeneity of the mixture) and an increase in the polyfractional character of the mixture. This is due to the very fine and monofractional composition of the original NCB 1450 sand.

According to these values, it can be assumed that, apart from BG 27 100%, the mixes containing 10–30% of all NCB grades will be the most likely to have stress defects overall. The addition of 10% NCB will not significantly affect the resulting dilatation and thus represents an inefficient choice.

Measured dilatation and sieve analysis results show a significant relationship between monofraction sands and dilatation rates. In general, the highest dilatation values are obtained for monofraction sands with round grains. This statement was confirmed in the case of the observed dilations for NCB 950 and 1450 sands (Figure 5a), which both have a very fine granulometric composition with a mean grain size d_50_ of 0.13–0.14 mm and a round grain shape. However, NCB 1450 is a sand with a more monofractional character compared to NCB 950, as mentioned in Section 3.1, where almost 95% of all grains are concentrated in two fractions. Despite being finer, NCB 1450 exhibits much higher final dilatations, by 27.8%. This is due to the fact that the monofractional composition of the sand is characterised by a much higher porosity even at high compaction levels, as it does not contain fine particles filling the intergranular spaces and, in this case, compensating for the dilatation of the larger grains. Therefore, in the case of more complex cores for the production of castings, where subsequent repair of the resulting stress defects would be impossible or very difficult, it is preferable to choose a sand of a more polyfractional character and avoid monofractional and sharply graded sands. Where the level of risk of defects is adequate or where defects can be easily removed by technological treatment, monofractional sands have the advantage of lower binder consumption due to their smaller specific surface area compared to polyfractional basic sands. Thus, it has been confirmed that, despite having the same d_50_ value and grain shape, monofraction basic sands achieve different dilatations and may be more vulnerable to stress defects such as veining. The highest dilatation values of the NCB sands in the initial condition were achieved by the NCB 650 basic sand at 0.56%. Here again, the dilatation result is consistent with the sieve analysis results, as it was the NCB sand with the largest mean grain size d_50_ of 0.19 mm. Here, the resulting dilatation is due to the larger grain size of the sand, despite the granulometric composition containing more fractions.

## 4. Conclusions

Several hybrid mixtures with silica sand and different types and contents of NCB sand were tested. The premise was to reduce the resulting dilatation and thermal expansion in the area of α-SiO_2_ phase transition. Investigated powder bulk density values demonstrate that the sands are very suitable to be combined without sedimenting, thus securing same properties of the mould or the core. A significant influence of mixing sands with different mean grain size d_50_ and the character of the fraction distribution on the final granulometric composition of the sands and its influence on the overall dilatation was confirmed. The highest reduction in dilatation of 32.2% was achieved by the addition of NCB 1450 50%. On the other hand, this mixture showed a higher increase in thermal expansion in the phase transition region, where a possible higher stress can be expected. The best result was achieved with the NCB 650 20% mixture, by 14.1% reduction in dilatation, when the other NCB types reached similar values starting at the 30% NCB content. Thus, the effect of the granulometric composition was confirmed, where the addition of NCB 650 increased the homogeneity of the hybrid mixture while maintaining the mean grain size d_50_ and more linear dilatation pattern. It has also been confirmed for 100% NCB sands that monofractions achieve different dilatations and may be more vulnerable to stress defects such as veining.

## Figures and Tables

**Figure 1 materials-15-06180-f001:**
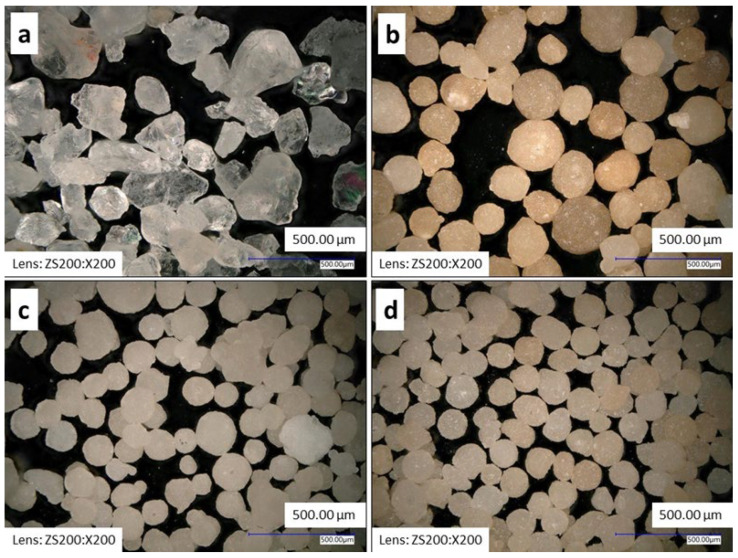
Detailed view of the sands used captured on a Keyence VHX 6000 digital microscope: (**a**) silica sand BG 27, (**b**) NCB 650, (**c**) NCB 950, (**d**) NCB 1450.

**Figure 2 materials-15-06180-f002:**
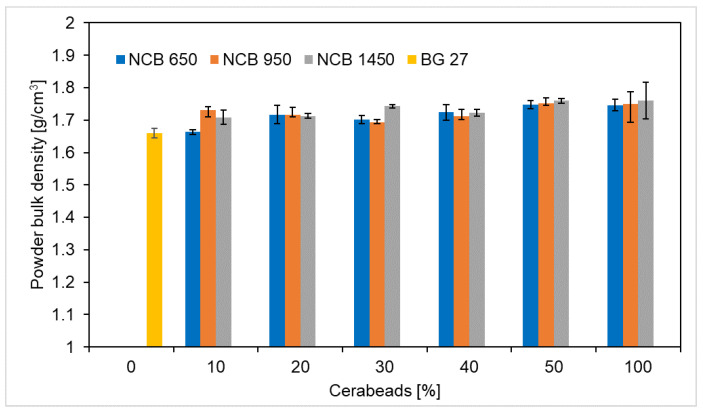
The trend of bulk density of hybrid mixtures as a function of CERABEADS content and its mean grain size.

**Figure 3 materials-15-06180-f003:**
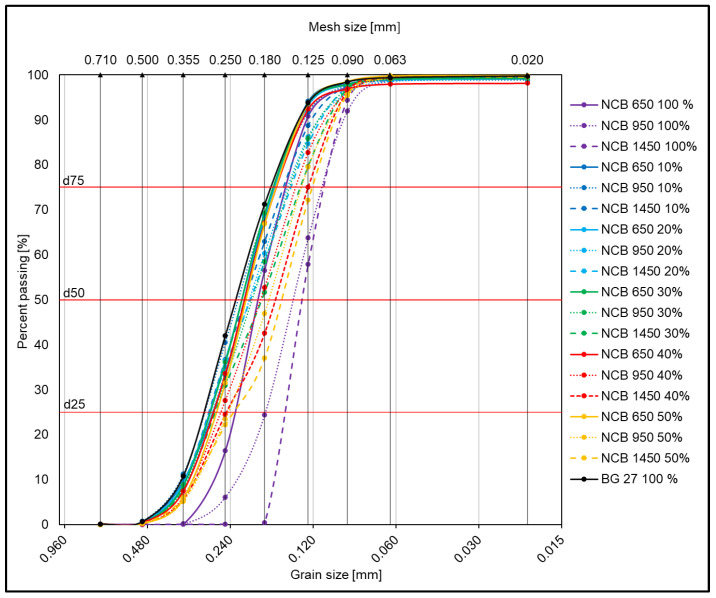
Cumulative curves of granularity for all pure and hybrid sands with various NCB contents.

**Figure 4 materials-15-06180-f004:**
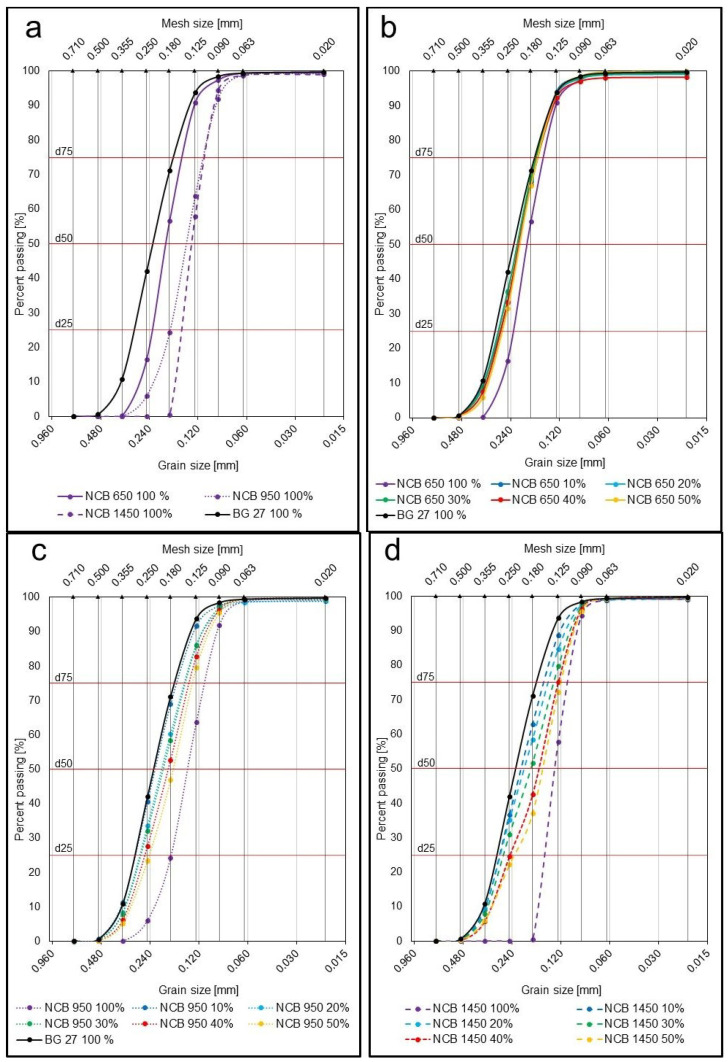
Detailed cumulative curves of granularity for individual NCB types compared with BG 27 100%: (**a**) 100% content of all sands. (**b**) NCB 650 addition. (**c**) NCB 950 addition. (**d**) NCB 1450 addition.

**Figure 5 materials-15-06180-f005:**
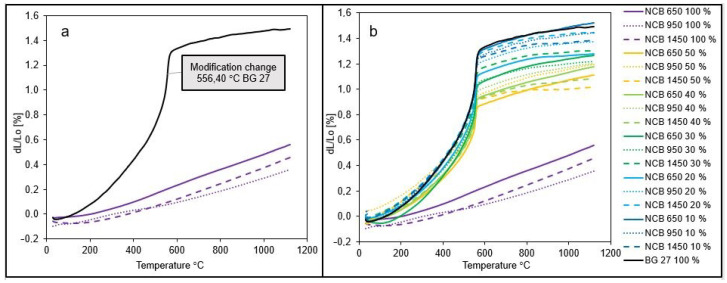
Dilatation curves for: (**a**) original silica and NCB sands in 100% content, (**b**) all tested pure and hybrid sand mixtures with various NCB contents.

**Figure 6 materials-15-06180-f006:**
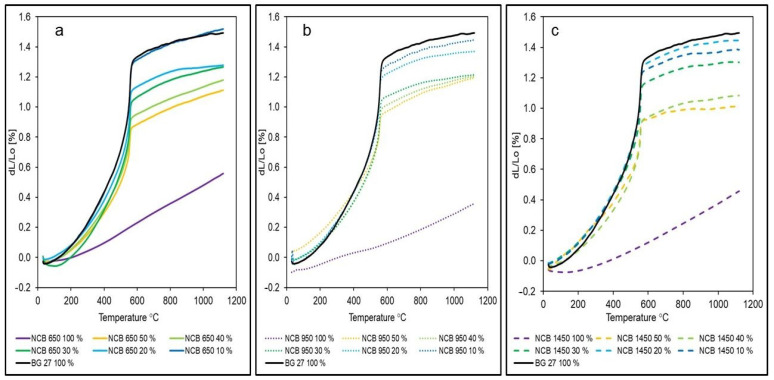
Detailed dilatation curves for individual NCB types compared with BG 27 100%: (**a**) NCB 650 hybrid sands, (**b**) NCB 950 hybrid sands, (**c**) NCB 1450 hybrid sands.

**Figure 7 materials-15-06180-f007:**
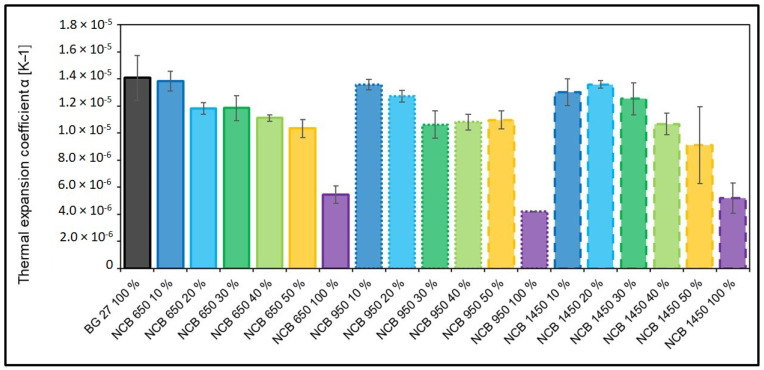
Comparison of thermal expansion coefficient of tested hybrid sands.

**Table 1 materials-15-06180-t001:** Sample marking and amounts of individual components.

Sample Marking	Amount of Silica Sand [%]	Amount of Non-Silica Sand [%]
	Biala Góra BG 27	CERABEADS
**BG 27 100%**	100	0
**NCB 650 10%**	90	10
**NCB 650 20%**	80	20
**NCB 650 30%**	70	30
**NCB 650 40%**	60	40
**NCB 650 50%**	50	50
**NCB 650 100%**	0	100
**NCB 950 10%**	90	10
**NCB 950 20%**	80	20
**NCB 950 30%**	70	30
**NCB 950 40%**	60	40
**NCB 950 50%**	50	50
**NCB 950 100%**	0	100
**NCB 1450 10%**	90	10
**NCB 1450 20%**	80	20
**NCB 1450 30%**	70	30
**NCB 1450 40%**	60	40
**NCB 1450 50%**	50	50
**NCB 1450 100%**	0	100

**Table 2 materials-15-06180-t002:** Particle size distribution and basic parameters of test sand specimens for NCB 650.

	Retained [%]						
Mesh Size [mm]	BG 27 100%	NCB 650 10%	NCB 650 20%	NCB 650 30%	NCB 650 40%	NCB 650 50%	NCB 650 100%
**0.710**	0.04	0.05	0.02	0.03	0.11	0.01	0.05
**0.500**	0.60	0.39	0.43	0.36	0.35	0.25	0.04
**0.355**	10.19	9.27	8.73	8.42	7.04	5.60	0.07
**0.250**	31.17	23.54	27.41	27.31	26.05	25.69	16.30
**0.180**	29.22	34.90	31.96	33.16	33.51	35.28	40.11
**0.125**	22.55	25.97	24.68	24.37	25.33	26.90	34.29
**0.090**	4.65	3.90	4.64	4.18	4.53	4.82	6.44
**0.063**	0.97	0.99	1.10	1.19	1.05	1.23	1.98
**Pan**	0.34	0.19	0.20	0.18	0.19	0.09	0.10
**AFS number**	61	62	62	62	62	64	70
**d_25_ [mm]**	0.30	0.28	0.29	0.28	0.28	0.27	0.23
**d_50_ [mm]**	0.23	0.21	0.22	0.22	0.21	0.21	0.19
**d_75_ [mm]**	0.17	0.17	0.17	0.17	0.16	0.16	0.15
**Sorting coefficient S [%]**	57.6	59.8	58.3	59.2	59.1	60.8	65.7
**Log W [-]**	64.0	63.3	65.7	64.6	64.2	62.9	56.3

**Table 3 materials-15-06180-t003:** Particle size distribution and basic parameters of test sand specimens for NCB 950.

	Retained [%]						
Mesh Size [mm]	BG 27 100%	NCB 950 10%	NCB 950 20%	NCB 950 30%	NCB 950 40%	NCB 950 50%	NCB 950 100%
**0.710**	0.04	0.01	0.01	0.00	0.01	0.01	0.01
**0.500**	0.60	0.77	0.35	0.31	0.11	0.16	0.01
**0.355**	10.19	10.49	8.04	7.74	6.07	5.03	0.07
**0.250**	31.17	29.25	25.20	24.12	21.44	18.27	5.96
**0.180**	29.22	28.42	26.70	26.33	25.12	23.49	18.31
**0.125**	22.55	22.78	25.93	27.39	30.04	32.55	39.37
**0.090**	4.65	5.86	9.85	11.07	13.49	15.95	28.23
**0.063**	0.97	1.10	2.44	2.30	3.01	3.80	6.73
**Pan**	0.34	0.15	0.63	0.23	0.39	0.47	0.56
**AFS number**	61	62	68	69	73	77	92
**d_25_ [mm]**	0.30	0.30	0.28	0.28	0.26	0.25	0.18
**d_50_ [mm]**	0.23	0.22	0.21	0.20	0.19	0.17	0.14
**d_75_ [mm]**	0.17	0.17	0.15	0.15	0.14	0.13	0.11
**Sorting coefficient S [%]**	57.6	55.4	53.4	53.3	53.5	54.3	62.7
**Log W [-]**	64.0	66.5	69.3	69.7	66.9	70.1	60.5

**Table 4 materials-15-06180-t004:** Particle size distribution and basic parameters of test sand specimens for NCB 1450.

	Retained [%]						
Mesh Size [mm]	BG 27 100%	NCB 1450 10%	NCB 1450 20%	NCB 1450 30%	NCB 1450 40%	NCB 1450 50%	NCB 1450 100%
**0.710**	0.04	0.01	0.07	0.02	0.01	0.01	0.01
**0.500**	0.60	0.34	0.45	0.27	0.17	0.25	0.03
**0.355**	10.19	8.83	8.97	7.53	5.49	5.77	0.02
**0.250**	31.17	27.52	25.61	23.10	18.87	16.15	0.05
**0.180**	29.22	26.24	23.40	20.72	18.01	14.86	0.31
**0.125**	22.55	25.83	26.21	28.06	32.52	35.12	57.50
**0.090**	4.65	8.95	12.76	17.34	21.27	23.56	36.53
**0.063**	0.97	1.15	1.82	2.26	3.05	3.61	4.40
**Pan**	0.34	0.31	0.21	0.22	0.37	0.20	0.24
**AFS number**	61	65	69	73	79	81	98
**d_25_ [mm]**	0.30	0.29	0.29	0.27	0.25	0.23	0.15
**d_50_ [mm]**	0.23	0.21	0.20	0.19	0.16	0.16	0.13
**d_75_ [mm]**	0.17	0.15	0.15	0.13	0.13	0.12	0.11
**Sorting coefficient S [%]**	57.6	53.5	50.9	49.3	50.7	51.7	72.4
**Log W [-]**	64.0	68.3	70.7	70.9	69.9	69.3	37.0

**Table 5 materials-15-06180-t005:** Comparison of thermal expansion coefficient of tested hybrid sands: results.

Hybrid Mixture	Thermal Expansion Coefficient α [K^−1^]	Standard Deviation Sx
**BG 27 100%**	1.407 × 10^−5^	1.636 × 10^−6^
**NCB 650 10%**	1.383 × 10^−5^	7.377 × 10^−7^
**NCB 650 20%**	1.181 × 10^−5^	4.350 × 10^−7^
**NCB 650 30%**	1.185 × 10^−5^	9.313 × 10^−7^
**NCB 650 40%**	1.111 × 10^−5^	2.285 × 10^−7^
**NCB 650 50%**	1.034 × 10^−5^	6.619 × 10^−7^
**NCB 650 100%**	5.446 × 10^−6^	6.406 × 10^−7^
**NCB 950 10%**	1.358 × 10^−5^	3.871 × 10^−7^
**NCB 950 20%**	1.270 × 10^−5^	4.249 × 10^−7^
**NCB 950 30%**	1.063 × 10^−5^	1.009 × 10^−6^
**NCB 950 40%**	1.081 × 10^−5^	5.741 × 10^−7^
**NCB 950 50%**	1.097 × 10^−5^	6.663 × 10^−7^
**NCB 950 100%**	4.202 × 10^−6^	1.575 × 10^−8^
**NCB 1450 10%**	1.302 × 10^−5^	1.005 × 10^−6^
**NCB 1450 20%**	1.359 × 10^−5^	2.799 × 10^−7^
**NCB 1450 30%**	1.254 × 10^−5^	1.177 × 10^−6^
**NCB 1450 40%**	1.067 × 10^−5^	7.854 × 10^−7^
**NCB 1450 50%**	9.112 × 10^−6^	2.851 × 10^−6^
**NCB 1450 100%**	5.192 × 10^−6^	1.099 × 10^−6^

## Data Availability

Not applicable.

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
