# Peer review of "Effect of the Composition of Hybrid Sands on the Change in Thermal Expansion"

_materials, 2022, doi:10.3390/ma15176180_

Round 1

Reviewer 1 Report

This paper presents a study on the gradation and thermal expansion property of various sands. Comments are provided below for authors’ consideration. 

L157: Why does a similar bulk density ensure good blendability? Please explain how the density affect blendability?

What is the size of each sand? What is the gradation?

What is the uncompacted air void of each sand? Angularity? Water absorption? Bulk specific gravity?

L175: What is BG and NCB?

L186: What ASTM procedure is followed for measuring bulk density?

L221: For thermal dilatation test of sands, is sand compacted or uncompacted? Do the voids among sand particles affect the result? How to control each specimen is consistent? What parameter is used to control the consistency of testing for different specimens?

Figures 4 is redundant, and also blurred. You have Tables 2 to 3 plus Figure 3, to show gradation.

L352 to L369: This discussion should be placed after introduction of results in Figure 5.

L354: There is an unusual symbol between β and α. What does it mean? The same in L395.

L371: What is the thermal expansion value exactly? Is it coefficient of thermal expansion? How is it measured? Is it one-directional or two-directional or volume change? In Figure 5, the y-axis is unclear.

Statistical analysis should be conducted for Figure 7 when comparing different materials.

The authors mentioned cost. However, there is no cost analysis to determine the optimal dosage in terms of thermal expansion property and percentage of NCB sand.

Does grain size affect dilatation result?

Reviewer 2 Report

Overall Comment

The paper entitled “Effect of the composition of hybrid sands on the change in thermal expansion”, showed some promising findings. Nonetheless, authors need to address some comments before ready for acceptance.

Abstract

Abstract focused too much on the background of the research. Reduce the background writeup and share some of the key results based on the thermal expansion results in the abstract. Include a concluding statement at the end.

Introduction

Introduction is well-written except for some minor typographical errors. Avoid using “we” in a scientific writeup, ideally, throughout the manuscript

Methodology

Generally, the work described in this section would be in past tense, unless theory discussion. Therefore, try to ensure this is consistent.

For section 2.1, from line 145 onwards, the writeup including the figure can be moved into the results and discussion.

Section 2.2, the information for CERABEADS should be more comprehensive.

For equation 1, 2a, and 2b, the information should be accompanied by ref.

Results and Discussion

General comment – avoid the usage of “us”, “we”.

Line 236 to 237, should be provided ref, as a support. If author did the density measurement, please include into the methodology.

Based on Figure 2, author included error bar in the graph. Please also indicate how author determine the error bar value in the methodology

Line 307, Line 332 – 334, need to be supported by literature

Line 335 -336 is not clear, the explanation should be more comprehensive

Figure 4 resolution need to be improved.

Line 382, phase transition was due to….?

Overall, the discussion is clear and good, however, some statements were not supported.

Conclusion

Should be more concise and how the research objectives has been met. Try to conclude in one paragraph, and again try to avoid “us”

Round 2

Reviewer 1 Report

The authors have addressed most comments, but did not conduct a statistical analysis for comparing different sands in Figure 7. Please consider statistical analysis to enhance this paper.
